**Data Availability Statement:** Data cannot be shared publicly owing to restrictions by law. Data

# Risk of placenta previa in assisted reproductive technology: A Nordic population study with sibling analyses

Eirik Landsverk[1]*, Kjersti Westvik-Johari[1,2], Ulla-Britt Wennerholm[3], Christina Bergh[3], Frederik Kyhl[4], Anne Lærke Spangmose[4], Ditte Vassard[4], Anja Pinborg[4,5], Kristiina Rönö[6], Mika Gissler[7,8], Sindre Hoff Petersen[1], Signe Opdahl[1]

1 Department of Public Health and Nursing, Faculty of Medicine and Health Science, Norwegian University of Science and Technology, Trondheim, Norway, 2 Department of Fertility, Division of Obstetrics and Gynecology, St. Olavs Hospital, Trondheim University Hospital, Trondheim, Norway, 3 Department of Obstetrics and Gynecology, Institute of Clinical Sciences, Sahlgrenska Academy, University of Gothenburg, Sahlgrenska University Hospital, Gothenburg, Sweden, 4 Fertility Clinic, Department of Gynaecology, Fertility and Obstetrics, Copenhagen University Hospital—Rigshospitalet, Copenhagen, Denmark, 5 Department of Clinical Medicine, University of Copenhagen, Copenhagen, Denmark, 6 Department of Obstetrics and Gynecology, University of Helsinki and Helsinki University Hospital, Helsinki, Finland, 7 Department of Data and Analytics, THL Finnish Institute for Health and Welfare, Helsinki, Finland, 8 Department of Molecular Medicine and Surgery, Karolinska Institutet, Stockholm, Sweden

* eirik.landsverk@ntnu.no

## Abstract

### Background

A higher risk of placenta previa after assisted reproductive technology (ART) is well established. The underlying mechanisms are poorly understood, but may relate to embryo culture duration, cryopreservation, and cause of infertility. Within-mother analyses, where each woman is her own control (i.e., sibling design), help disentangle treatment contributions from maternal confounders that are stable between pregnancies. We aimed to investigate the risk of placenta previa in pregnancies achieved after ART according to embryo culture duration, cryopreservation, and infertility factors while accounting for stable maternal factors using within-mother analyses.

### Methods and findings

We used linked nationwide registry data from Denmark (1994 to 2014), Finland (1990 to 2014), Norway (1988 to 2015), and Sweden (1988 to 2015). All women who gave their first birth during the study period at age 20 years or older were eligible and contributed up to 4 deliveries (singleton or multifetal) occurring between 22 and 44 weeks of gestation, excluding deliveries where maternal age exceeded 45 years. We used multilevel logistic regression to compare risk of placenta previa after ART ($n$ = 139,694 deliveries) versus natural conception ($n$ = 5,614,512 deliveries), both at the population level and within mothers, adjusting for year of delivery, maternal age, parity, and country. We categorized ART according to culture duration, embryo cryopreservation, and infertility factors. Population

are available from the CoNARTaS server at Statistics Denmark, after approval by the Ethics Committees and registry keeping authorities in each country, as described in the following publication: Opdahl S, Henningsen AA, Bergh C, Gissler M, Romundstad LB, Petzold M, Tiitinen A, Wennerholm UB, Pinborg AB. Data Resource Profile: Committee of Nordic Assisted Reproductive Technology and Safety (CoNARTaS) cohort. Int J Epidemiol. 2020 Apr 1;49(2):365-366f. doi: 10.1093/ije/dyz228. Contact information for Statistics Denmark: Division of Research Services, Statistics Denmark, Sejrøgade 11, DK-2100, Copenhagen, Denmark. E-mail: forskningsservice@dst.dk. Phone: +45 39 17 31 30.

**Funding:** This work was supported by NordForsk (71450 to SO), the Central Norway Regional Health Authorities (46045000 to SO), the Nordic Federation of Obstetrics and Gynaecology (NF13041, NF15058, NF16026 and NF17043 to UBW), and the Interreg Öresund-Kattegat-Skagerrak European Regional Development Fund (ReproUnion project, to AP & CB). The funders had no role in the study design, data collection and analysis, decision to publish, or preparation of the manuscript.

**Competing interests:** AP declares grants from Gedeon Richter, Ferring, Cryos, and Merck, consulting fees from IBSA, Ferring, Gedeon Richter, Cryos, and Merck, payments from Gedeon Richter, Ferring, Merck, and Organon, travel support from Gedeon Richter. CB declares grants from Ferring, payments from Ferring, and travel support from Gedeon Richter. All other authors of this paper have no conflicts of interest to declare. None of the companies listed had any financial stake in the results of this study.

**Abbreviations:** aOR, adjusted odds ratio; ART, assisted reproductive technology; BT, blastocyst transfer; CI, confidence interval; CT, cleavage stage embryo transfer; ET, embryo transfer; IVF, in vitro fertilization; MBR, Medical Birth Registry; NC, natural conception; OR, odds ratio; PCOS, polycystic ovary syndrome.

level risk of placenta previa was higher for ART versus natural conception (odds ratio [OR], 4.16; 95% confidence interval [CI], 3.96–4.37). Controlling for stable maternal factors, the association attenuated, but risk remained higher for ART versus natural conception (OR within mothers, 2.64; 95% CI, 2.31–3.02). Compared to naturally conceived, a larger difference in risk was seen for pregnancies from fresh embryos than for pregnancies from frozen embryos. Further categorization by culture duration showed the largest risk difference after fresh blastocyst transfer, and the smallest after frozen cleavage stage embryo transfer, which persisted in sensitivity analyses (including restriction to singletons). When stratified according to infertility factors at the population level, women with endometriosis conceiving by ART had the highest risk of placenta previa (OR, 9.35; 95% CI, 8.50–10.29), whereas women with polycystic ovary syndrome (PCOS) conceiving by ART had the lowest risk (OR, 1.52; 95% CI, 1.12–2.09), compared to natural conception. Within mothers, we found a higher risk of placenta previa after ART compared to natural conception for women with endometriosis (OR, 2.08; 95% CI, 1.50–2.90), but not for women with PCOS (OR, 0.88; 95% CI, 0.41–1.89 [unadjusted due to sparse data]). However, within-mother analyses are restricted to multiparous women with deliveries after different conception methods. Therefore, findings from these analyses might not generalize to all women undergoing ART.

## Conclusions

The risk of placenta previa in pregnancies conceived by ART differed by embryo culture duration, cryopreservation, and underlying infertility. The highest risk was seen after fresh embryo transfer and especially fresh blastocyst transfer. Women with endometriosis had a higher risk than women with other infertility factors, and within mothers, their risk was higher after ART than after natural conception. Identifying the responsible mechanisms might provide opportunities for prevention.

## Author summary

### Why was this study done?

- Studies indicate risk of placenta previa in assisted reproductive technology (ART) differs by type of ART method and underlying infertility.

- We aimed to increase our understanding of these factors' contributions by improved confounding control using within-mother (sibling) analyses.

### What did the researchers do and find?

- Using each mother as her own control, we found the highest risk of placenta previa after fresh blastocyst transfer.

- In addition, we found a markedly higher risk among women with endometriosis who conceived after assisted reproductive technology, which remained doubled compared to natural conception after controlling for maternal factors through within-mother analyses.

### What do these findings mean?

- This study strengthens evidence that treatment and maternal factors may both contribute to risk of placenta previa in assisted reproductive technology.

- Further insights into the mechanisms responsible might lead to development of safer ART methods.

- Although within-mother analyses are less prone to confounding from stable maternal factors, the findings of these analyses require that women have several births; hence, results might not apply to all women using ART.

## Introduction

The use of assisted reproductive technology (ART) is increasing worldwide [1]. This can be attributed to increasing success rates and availability of ART, and postponement of parenthood to an age with lower fecundity [2,3]. Compared to natural conception (NC), ART is associated with higher risk of pregnancy complications such as hypertensive disorder of pregnancy, gestational diabetes mellitus, and placenta previa [4–6], as well as adverse perinatal outcomes including small for gestational age, preterm birth, and perinatal death [7].

In Europe, more than 200,000 children are born after ART each year (constituting 3% of the total birth cohorts), and transfer of single embryos has greatly reduced multifetal gestations and hence perinatal complications in ART-conceived pregnancies [3,8,9]. In contrast, the risk of placenta previa after ART has increased over time in the Nordic countries [10], which is concerning as placenta previa is associated with complications for both the mother and child [11–17].

Previous studies have linked both fresh embryo transfer (fresh-ET) and blastocyst transfer, and the combination of the two, with increased risk of placenta previa [18–26], but these findings are inconsistent [27,28]. Differences in underlying infertility might explain some of the discrepancies between previous studies [18,29,30]. Separating the contributions from underlying infertility and ART may be possible with a within-mother design, i.e., sibling design [31,32].

In a large study population, we investigated the risk of placenta previa according to embryo culture duration (i.e., cleavage stage embryo transfer versus blastocyst stage embryo transfer), cryopreservation, and infertility factors, accounting for maternal factors using within-mother analyses.

## Materials and methods

This study is reported as per the Reporting of Studies Conducted using Observational Routinely-Collected Data (RECORD) guideline (S1 Checklist). All analyses were planned a priori as outlined in S1 Study Protocol.

### Data sources and variables

We used data from the Committee of Nordic Assisted Reproductive Technology and Safety (CoNARTaS) cohort, which comprises all deliveries in the Medical Birth Registries (MBRs) in Denmark, Finland, Norway, and Sweden since the start of national ART registration. The

MBRs collect information on all deliveries in each country. The mothers' unique national identity number was pseudonymized before individual data linkage between the MBRs and other national health registries, described in more detail in a previous publication by our research group [33].

The exposure was conception by ART, defined as fertilization outside the female body, i.e., fertilization with in vitro fertilization (IVF) or intracytoplasmic sperm injection [34]. ART-conception was identified through ART registries in Denmark and Sweden, and MBRs in Norway and Finland [33]. Pregnancies after ovulation induction and insemination were categorized as naturally conceived together with pregnancies not registered as ART-conceived. Danish, Norwegian, and Swedish ART data could be further categorized as cleavage stage embryo transfer (culture duration 2 to 3 days) or blastocyst transfer (culture duration 5 to 6 days), and as fresh-ET or frozen thawed embryo transfer (frozen-ET).

The outcome was placenta previa, identified from the MBRs in Finland, Norway, and Sweden, and the National Patient Registries of Denmark and Finland (S1 Table). Using these registries, we could not differentiate between total, partial, or marginal placenta previa, or low-lying placenta (<2 cm from the internal os). We only included diagnoses during the third trimester or within 1 month before delivery from the National Patient Registries because most low-lying placentas in the second trimester resolve in the third trimester [35]. In the Nordic countries, placenta previa is usually discovered at second trimester routine ultrasonography, and follow-up is normally performed in week 32 to 34 to evaluate whether it has resolved [36–38].

Infertility factors (i.e., diagnoses associated with infertility) were identified using the National Patient Registries of each country, supplemented with information from the national IVF registry in Denmark, and the Medical Birth Registry of Norway (S2 and S3 Tables). We assigned each woman to only one of these factors during the study period. Factors more strongly associated with the risk of placenta previa [18,29,30], and factors with low prevalence, were given priority in this order: uterine factors, endometriosis, polycystic ovary syndrome (PCOS), male-factor infertility only, and other infertility factors (including unspecified female infertility and unexplained infertility).

## Study population

The study period was defined as 1994 to 2014 in Denmark, 1990 to 2014 in Finland, and 1988 to 2015 in Norway and Sweden. Only deliveries by women whose first delivery occurred during the study period at age ≥20 years were defined as eligible (Fig 1). This was done to ensure sufficient overlap in year of delivery, maternal age, and parity between ART and NC [39]. For these women, up to the first 4 deliveries were included (parity 0–3), excluding deliveries at maternal age >45 years. In addition, deliveries with extreme (<22 or >44 weeks) or missing gestational age were excluded. We included multifetal pregnancies because this does not seem to be a clear risk factor of placenta previa [9,10,30,40]. ART-conceived pregnancies after gamete donation and/or with preimplantation genetic testing were also included. **Sample 1** included 5,754,206 deliveries in all 4 countries combined. Among these, 139,694 were from ART-conceived pregnancies, and 42,792 women gave birth after both ART and NC during the study period (Fig 1).

In **sample 2**, ART-conceived pregnancies with missing cryopreservation status were excluded. To create **sample 3**, ART-conceived pregnancies with missing culture duration were excluded, leaving 94,631 deliveries after ART for analyses on combinations of culture duration and cryopreservation (Fig 1). We also excluded pregnancies after NC from Finland in samples 2 and 3 as there was no information on culture duration or cryopreservation among ART-conceived pregnancies in Finland.

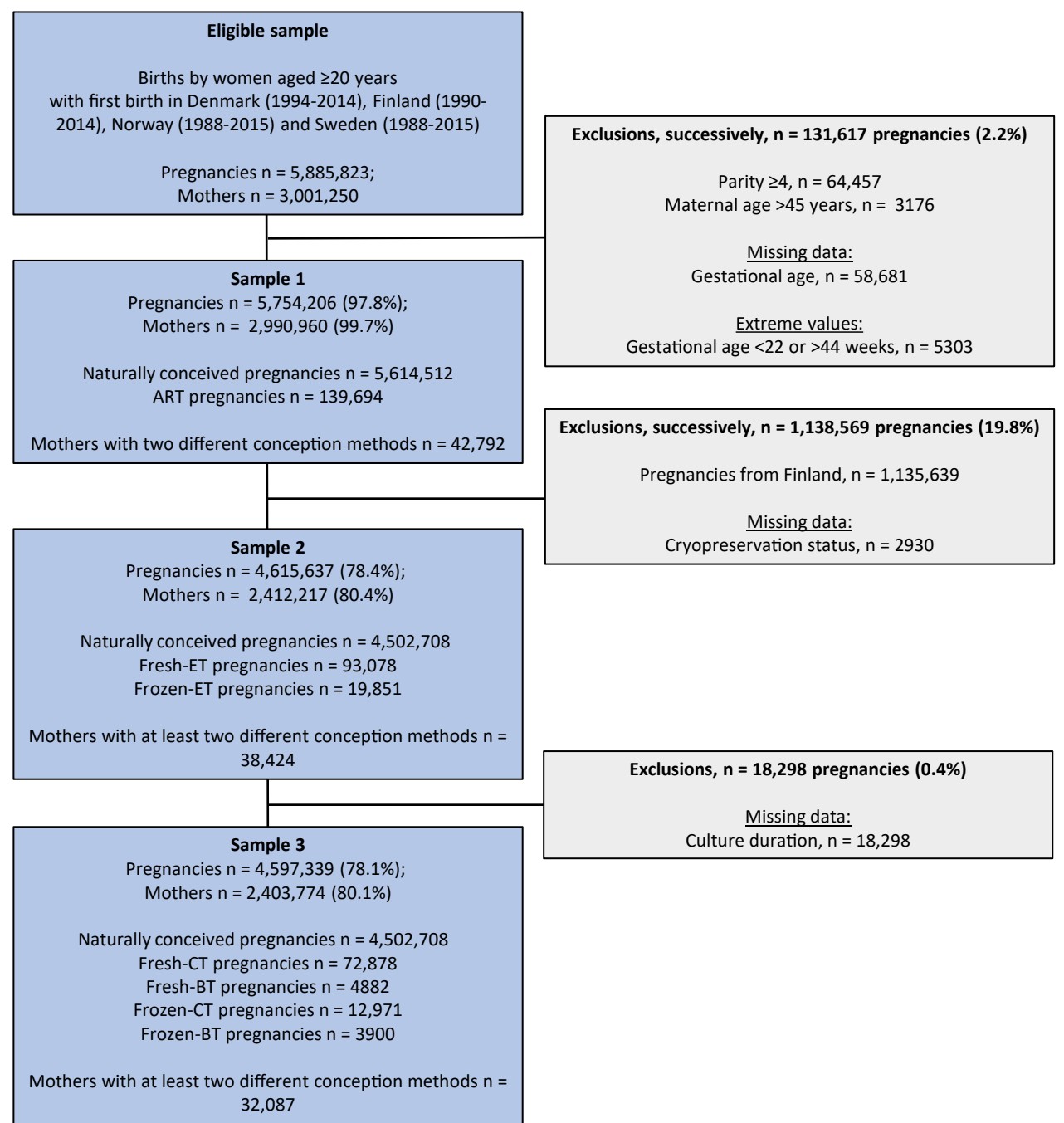

**Fig 1. Flow chart of study population showing eligibility, exclusion criteria and numbers, and number of exposed and unexposed in the 3 analyses samples.** ART, assisted reproductive technology; fresh-ET, fresh embryo transfer; frozen-ET, frozen embryo transfer; fresh-CT, fresh cleavage stage embryo transfer; fresh-BT, fresh blastocyst transfer; frozen-CT, frozen cleavage stage embryo transfer; frozen-BT, frozen blastocyst transfer.

## Statistical analyses

We estimated adjusted odds ratios (aORs) for placenta previa at the population level and within mothers using multilevel logistic regression with deliveries nested within mothers

(using the "xtlogit" command in Stata with maternal identity codes specified as clusters). This hierarchical approach adjusts for clustering of pregnancies by the same mother. Population-level associations were estimated in random intercept models using the full analysis samples, where each mother contributed 1 to 4 deliveries (multifetal pregnancy counting as 1 delivery). Within mother associations were estimated using a fixed intercept model [41], where each mother contributed 2 to 4 deliveries. In the latter model, deliveries contribute to the final exposure-outcome estimate only if both the exposure and outcome differ between pregnancies by the same mother. As a sensitivity analysis, we restricted both models to only include the first 2 consecutive deliveries by each mother. Available data on factors known to influence the probability of ART conception and the risk of placenta previa were included as covariates in the regression models; year of delivery (categorical: 1988–1996, 1997–2001, 2002–2006, 2007–2011, 2012–2015), maternal age (categorical: 20–24, 25–29, 30–34, 35–39, 40–45), parity (categorical: 0, 1, 2, 3), and country.

We used all NC pregnancies as the reference in population level analyses of infertility factors. Within mothers, we performed separate analyses for each infertility factor, restricting analyses to women assigned to that factor.

In sample 2, we examined whether risk of placenta previa within mothers was driven by certain combinations of parity and conception method with ART categorized as either fresh-ET or frozen-ET. Specifically, we repeated the multilevel logistic regression at the population level with an interaction term between parity and the resulting 9 combinations of conception methods for the mothers' first and second delivery and obtained predicted risks of placenta previa using postestimation commands. We also did a series of other sensitivity analyses to test the robustness of the results, some of which are described in more detail in S1 Text.

Finally, to assess the clinical severity of placenta previa, we compared other pregnancy outcomes (gestational age, birthweight, and cesarean section) between deliveries with and without placenta previa according to conception method and plurality.

We used Stata 18 (Statacorp LLC, Texas, United States of America) for all statistical analyses.

## Ethics

In Norway, the study was approved by the Regional Committees for Medical and Health Research Ethics (REK 2010/1909). In Sweden, the study was approved by the Ethical committee in Gothenburg (Dnr 214–12, T422-12, T516-15, T233-16, T300-17, T1144-17, T121-18). In Denmark and Finland, ethical approval is not required for research only based on registry data. In accordance with legal regulations of the participating countries, written consent from participants is not required when using registry data in Denmark and Finland, and the ethical committees gave exemption from obtaining consent in Norway and Sweden.

## Results

Background characteristics differed between pregnancies after ART and NC (Table 1). The number of deliveries after ART increased over the study period. Women who conceived by ART were on average 4 years older, had lower parity and less frequently smoked during pregnancy, compared to naturally conceiving women. In addition, twin pregnancies were more common after ART than NC (14.0% versus 1.3%).

For ART versus NC at the population level, aOR of placenta previa was 4.16 (95% confidence interval [CI], 3.96–4.37). Comparisons within mothers who delivered after both ART and NC also showed a clearly higher risk after ART, although the point estimate was lower than at the population level (aOR, 2.64; 95% CI, 2.31–3.02). At the population level, risk was

**Table 1. Description of the study population in sample 1.**

| | ART | NC |
|---|---|---|
| Total number of pregnancies, *n* | 139,694 | 5,614,512 |
| Country, *n* (%) | | |
| Denmark | 35,365 (25.3) | 1,001,886 (17.8) |
| Finland | 23,835 (17.1) | 1,111,804 (19.8) |
| Norway | 26,805 (19.2) | 1,217,113 (21.7) |
| Sweden | 53,689 (38.4) | 2,283,709 (40.7) |
| Year of birth, *n* (%) | | |
| 1988–1996 | 11,624 (8.3) | 1,295,414 (23.1) |
| 1997–2001 | 22,371 (16.0) | 1,046,979 (18.7) |
| 2002–2006 | 30,251 (21.7) | 1,165,294 (20.8) |
| 2007–2011 | 41,727 (29.9) | 1,234,664 (22.0) |
| 2012–2015 | 33,721 (24.1) | 872,161 (15.5) |
| Mean maternal age (SD) | 33.7 (4.2) | 29.6 (4.8) |
| Parity, *n* (%) | | |
| 0 | 100,857 (72.2) | 2,862,509 (51.0) |
| 1 | 33,881 (24.3) | 1,998,172 (35.6) |
| 2 | 4,251 (3.0) | 620,801 (11.1) |
| 3 | 705 (0.5) | 133,030 (2.4) |
| Smoking[a], *n* (%) | | |
| Yes | 7,198 (5.3) | 610,606 (11.8) |
| No | 120,830 (89.2) | 4,280,132 (82.5) |
| Missing[b] | 7,362 (5.4) | 299,460 (5.8) |
| Maternal BMI[c] (kg/m$^2$), mean (SD) | 24.2 (4.1) | 24.2 (4.5) |
| Missing[b], *n* (%) | 14,210 (13.4) | 538,828 (14.5) |
| Infertility factors | | |
| Uterine factor[d] | 1,517 (1.1) | NA |
| Endometriosis[e] | 16,205 (11.6) | NA |
| PCOS[f] | 7,051 (5.1) | NA |
| Male factors only[g] | 10,222 (7.3) | NA |
| Other infertility factors[h] | 97,401 (70.0) | NA |
| Missing[i] | 7,298 (5.2) | NA |
| Plurality, *n* (%) | | |
| Singleton | 119,591 (85.6) | 5,540,830 (98.7) |
| Twin | 19,513 (14.0) | 72,504 (1.3) |
| Triplet or higher | 590 (0.4) | 1,178 (0.0) |

[a] Smoking was recorded from 1999 in Norway and prior to our study period in Denmark, Finland, and Sweden.

[b] Missing during registration period.

[c] BMI was recorded from 2007 in Norway, 2004 in Denmark and Finland, and prior to our study period in Sweden.

[d] Pregnancies by women once diagnosed with female infertility due to uterine factors.

[e] Pregnancies by women once diagnosed with endometriosis without any diagnosis of female infertility due to uterine factors.

[f] Pregnancies by women once diagnosed with polycystic ovary syndrome, without any diagnosis of female infertility due to uterine factors nor endometriosis.

[g] Pregnancies by women once diagnosed with infertility due to male factors, without any diagnosis of infertility due to female factors.

[h] Pregnancies by women with other infertility factors than uterine factors, endometriosis, PCOS, and male factors only. This category also includes pregnancies by women with unspecified female infertility and unexplained infertility.

[i] Pregnancies by women missing any diagnosis code associated with infertility.

ART, assisted reproductive technology; NC, natural conception, NA, not applicable; PCOS, polycystic ovary syndrome.

**Table 2. Observed risk and adjusted odds ratio of placenta previa at the population level and within mothers according to conception method and sample.**

|  | Population level | | | Within mothers | | |
|---|---|---|---|---|---|---|
|  | **N** | **Observed risk** | **Adjusted odds ratio[a] (95% CI)** | **N[b]** | **Observed risk[c]** | **Adjusted odds ratio[d] (95% CI)** |
| Sample 1 |  |  |  |  |  |  |
| NC | 5,614,512 | 3.1/1,000 | 1 (ref) | 51,521 | 7.3/1,000 | 1 (ref) |
| ART | 139,694 | 17.2/1,000 | 4.16 (3.96 to 4.37) | 44,843 | 17.8/1,000 | 2.64 (2.31 to 3.02) |
| Sample 2 |  |  |  |  |  |  |
| NC | 4,502,708 | 2.8/1,000 | 1 (ref) | 39,237 | 6.4/1,000 | 1 (ref) |
| Fresh-ET | 93,078 | 17.4/1,000 | 4.69 (4.41 to 4.98) | 34,810 | 18.3/1,000 | 3.31 (2.81 to 3.91) |
| Frozen-ET | 19,851 | 11.1/1,000 | 2.71 (2.35 to 3.13) | 10,724 | 10.4/1,000 | 1.59 (1.19 to 2.13) |
| Sample 3 |  |  |  |  |  |  |
| NC | 4,502,708 | 2.8/1,000 | 1 (ref) | 32,182 | 6.6/1,000 | 1 (ref) |
| Fresh-CT | 72,878 | 16.5/1,000 | 4.24 (3.96 to 4.54) | 27,497 | 17.3/1,000 | 3.07 (2.55 to 3.69) |
| Fresh-BT | 4,882 | 41.0/1,000 | 11.51 (9.75 to 13.57) | 2401 | 40.0/1,000 | 7.80 (4.59 to 13.27) |
| Frozen-CT | 12,971 | 9.7/1,000 | 2.33 (1.93 to 2.80) | 7303 | 7.9/1,000 | 1.20 (0.83 to 1.75) |
| Frozen-BT | 3,900 | 15.6/1,000 | 4.04 (3.07 to 5.31) | 1768 | 19.8/1,000 | 3.13 (1.71 to 5.73) |

[a] Adjusted for year of delivery (categorical: 1988–1996, 1997–2001, 2002–2006, 2007–2011, 2012–2015), maternal age (categorical: 20–24, 25–29, 30–34, 35–39, 40–45), parity (categorical: 0, 1, 2, 3), and country.

[b] Total number of pregnancies from mothers with at least 2 different conception methods.

[c] Observed risk in pregnancies from mothers with at least 2 different conception methods.

[d] Adjusted for year of delivery, maternal age, and parity (country is constant within mothers and therefore not included as a covariate in these analyses). Estimated by logistic regression (fixed intercept model) on all pregnancies in each sample.

ART, assisted reproductive technology; CI, confidence interval; fresh-ET, fresh embryo transfer; frozen-ET, frozen embryo transfer; fresh-CT, fresh cleavage stage embryo transfer; fresh-BT, fresh blastocyst transfer; frozen-CT, frozen cleavage stage embryo transfer; frozen-BT, frozen blastocyst transfer; NC, natural conception.

higher after fresh-ET (aOR, 4.69; 95% CI, 4.41–4.98) than after frozen-ET (aOR, 2.71; 95% CI, 2.35–3.13), using NC as reference. When further categorizing fresh-ET and frozen-ET according to culture duration, population-level risk was higher for all treatment combinations compared to NC (Table 2), with the highest risk after fresh blastocyst transfer (fresh-BT, aOR, 11.51; 95% CI, 9.75–13.57). Within mothers, risk estimates attenuated to some extent, but remained higher for all treatment combinations except frozen cleavage stage embryo transfer (frozen-CT, Table 2).

At the population level, ART-conception was associated with higher risk of placenta previa compared to NC for all infertility factors (Table 3), ranging from highest among women with endometriosis (aOR, 9.35; 95% CI, 8.50–10.29) to lowest among women with PCOS (aOR, 1.52; 95% CI, 1.12–2.09). Within mothers, the associations attenuated compared to the population level, but risk remained higher in ART versus NC for all infertility factors (Table 3), except for PCOS (odds ratio, 0.88; 95% CI, 0.41–1.89 [unadjusted due to sparse data]). For women with endometriosis, aOR was 2.08 (95% CI, 1.50–2.90) for ART versus NC within mothers. Giving endometriosis and PCOS first priority in the hierarchy of infertility factors, we found similar results both for endometriosis (population level aOR, 9.31; 95% CI, 8.47–10.24; aOR within mothers, 2.10; 95% CI, 1.51–2.92) and for PCOS (population-level aOR, 1.89; 95% CI, 1.44–2.48; OR within mothers, 0.82; 95% CI, 0.41–1.65 [unadjusted due to sparse data]).

The risk of placenta previa in the second delivery was highest after fresh-ET, intermediate after frozen-ET, and lowest after NC when stratified according to conception method in the first delivery (Fig 2). Notably, the risk increased between first and second delivery for all combinations of conception method, except when fresh-ET was followed by frozen-ET or NC, where the risk was lower in the second delivery compared to the first (Fig 2).

**Table 3. Observed risk and adjusted odds ratio of placenta previa at the population level and within mothers according to conception method and infertility factor in sample 1.**

| | Population level[a] | | | Within mothers[b] | | |
|---|---|---|---|---|---|---|
| | N | Observed risk | Adjusted odds ratio[c] (95% CI) | N[d] | Observed risk[e] | Adjusted odds ratio[f] (95% CI) |
| Uterine factors[g] | | | | | | |
| NC | 5,614,512 | 3.1/1,000 | 1 (ref) | 472 | 6.4/1,000 | 1 (ref) |
| ART | 1517 | 18.5/1,000 | 3.76 (2.51 to 5.65) | 430 | 20.9/1,000 | 3.06 (0.82 to 11.41)[h] |
| Endometriosis[i] | | | | | | |
| NC | 5,614,512 | 3.1/1,000 | 1 (ref) | 5481 | 15.5/1,000 | 1 (ref) |
| ART | 16,205 | 37.3/1,000 | 9.35 (8.50 to 10.29) | 4983 | 30.7/1,000 | 2.08 (1.50 to 2.90) |
| PCOS[j] | | | | | | |
| NC | 5,614,512 | 3.1/1,000 | 1 (ref) | 2434 | 6.6/1,000 | 1 (ref) |
| ART | 7,051 | 6.1/1,000 | 1.53 (1.12 to 2.09) | 2159 | 5.6/1,000 | 0.88 (0.41 to 1.89)[h] |
| Male factors only[k] | | | | | | |
| NC | 5,614,512 | 3.1/1,000 | 1 (ref) | 3532 | 4.8/1,000 | 1 (ref) |
| ART | 10,222 | 11.3/1,000 | 3.04 (2.50 to 3.70) | 3110 | 13.5/1,000 | 2.91 (1.64 to 5.17)[h] |
| Other infertility factors[l] | | | | | | |
| NC | 5,614,512 | 3.1/1,000 | 1 (ref) | 39,723 | 6.4/1,000 | 1 (ref) |
| ART | 104,699 | 15.4/1,000 | 3.49 (3.30 to 3.69) | 34,377 | 17.0/1,000 | 2.76 (2.31 to 3.29) |

[a] At the population level, infertility factor was only defined for pregnancies after ART.

[b] Within mothers, infertility factor was only defined for women with an ART conception during the study period.

[c] Adjusted for year of delivery (categorical: 1988–1996, 1997–2001, 2002–2006, 2007–2011, 2012–2015), maternal age (categorical: 20–24, 25–29, 30–34, 35–39, 40–45), parity (categorical: 0, 1, 2, 3), and country.

[d] Total number of pregnancies from mothers with pregnancies after both NC and ART during the study period.

[e] Observed risk in pregnancies from mothers with pregnancies after both NC and ART during the study period.

[f] Adjusted for year of delivery (categorical: 1988–1996, 1997–2001, 2002–2006, 2007–2011, 2012–2015), maternal age (categorical: 20–24, 25–29, 30–34, 35–39, 40–45), and parity (categorical: 0, 1, 2, 3). Country is constant within mothers and therefore not included as a covariate in these analyses. Estimated by logistic regression (fixed intercept model) on all pregnancies in each category of infertility factor.

[g] Pregnancies by women once diagnosed with female infertility due to uterine factors.

[h] The odds ratios comparing ART vs. NC within mothers in women with uterine factors, PCOS, and male factors only are unadjusted due to sparse data.

[i] Pregnancies by women once diagnosed with endometriosis without any diagnosis of female infertility due to uterine factors.

[j] Pregnancies by women once diagnosed with polycystic ovary syndrome, without any diagnosis of female infertility due to uterine factors nor endometriosis.

[k] Pregnancies by women once diagnosed with infertility due to male factors, without any diagnosis of infertility due to female factors.

[l] Pregnancies by women with other infertility factors than uterine factors, endometriosis, PCOS, and male factors only. This category also includes pregnancies by women with unspecified female infertility, unexplained infertility, and missing any diagnosis code associated with infertility.

ART, assisted reproductive technology; CI, confidence interval; NC, natural conception; PCOS, polycystic ovary syndrome.

All sensitivity analyses were consistent with the main results. When stratified on characteristics of the first pregnancy (conception method and cesarean section/placenta previa), risk of placenta previa in the second pregnancy was consistently higher in ART-conceived pregnancies compared to NC (S4 and S5 Tables). Associations between embryo culture duration, cryopreservation, and placenta previa were also consistent across sensitivity analyses (Fig 3). Although results for fresh-BT and frozen blastocyst transfer (frozen-BT) attenuated somewhat within mothers when restricting ART-conceived pregnancies to fertilization with IVF (i.e., without intracytoplasmic sperm injection), risk still tended to be higher after fresh-BT than after other combinations of culture duration and cryopreservation status in this subpopulation (Fig 3).

ART-conceived singleton pregnancies with placenta previa had a shorter mean gestational age (259.0 versus 276.9 days), a higher rate of preterm birth (36.3% versus 7.5%), and a lower

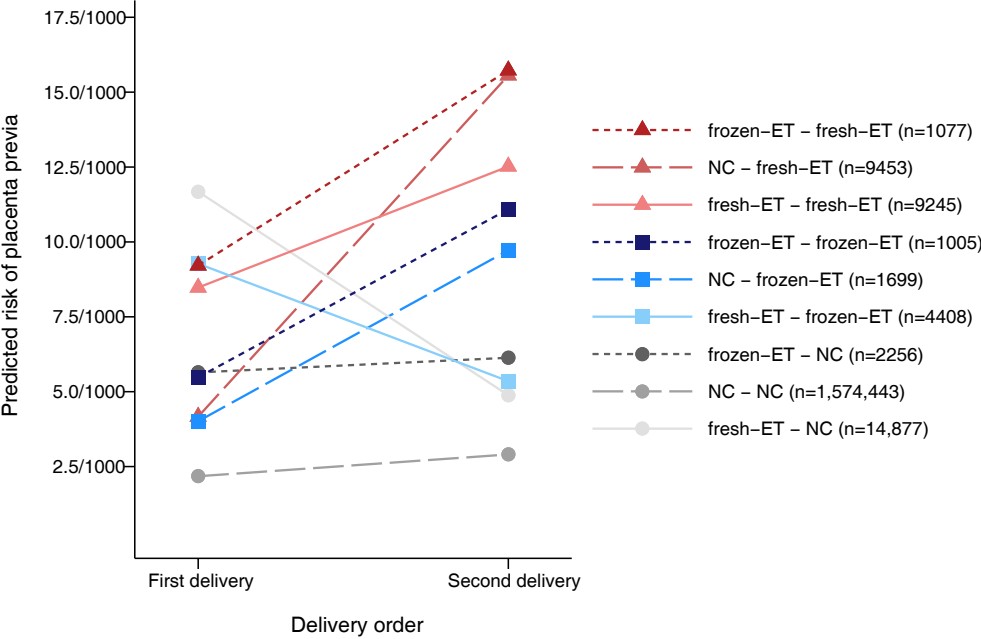

**Fig 2. Predicted risk of placenta previa in the 2 first deliveries according conception methods in sample 2.** Frozen-ET, frozen embryo transfer; fresh-ET, fresh embryo transfer; NC, natural conception. We calculated predicted risk of placenta previa using multilevel logistic regression at the population level and marginal standardization ("margins" command in Stata [StataCorp, College Station, TX]). We included the following covariates in the regression model: parity (0, 1), conception methods for the mothers' first and second delivery (categorical: NC–NC, NC–fresh-ET, NC–frozen-ET, fresh-ET–NC, fresh-ET–fresh-ET, fresh-ET–frozen-ET, frozen-ET–NC, frozen-ET–fresh-ET, frozen-ET–frozen-ET), year of delivery (categorical: 1988–1996, 1997–2001, 2002–2006, 2007–2011, 2012–2015), maternal age (categorical: 20–24, 25–29, 30–34, 35–39, 40–45), and country. We added an interaction term between parity and conception methods for the mothers' first and second delivery to allow the slope of the lines to differ.

mean birthweight (2,986 versus 3,447 g). However, they exhibited similar mean birthweight z-scores and comparable proportions of small and large for gestational age compared to ART-conceived singleton pregnancies without placenta previa (S6 Table). A similar trend of reduced birthweight and gestational age was observed in naturally conceived pregnancies with placenta previa compared to those without, for both singleton and twin pregnancies. This pattern was also evident in ART-conceived twin pregnancies with placenta previa compared to those without (S6 Table).

## Discussion

The increased risk of placenta previa in ART-conceived pregnancies differed by embryo culture duration and cryopreservation, both at the population level and within mothers, with a higher risk after fresh-ET, and especially fresh-BT. Estimates attenuated from population level to within mothers, suggesting stable maternal factors contribute to the increased risk. This was further supported by large differences in risk according to infertility factors, where women with ART-conception and endometriosis had the highest risk of placenta previa. Their risk after ART-conception remained higher in within-mother analyses, suggesting that stable maternal factors alone do not fully explain the increased risk associated with ART.

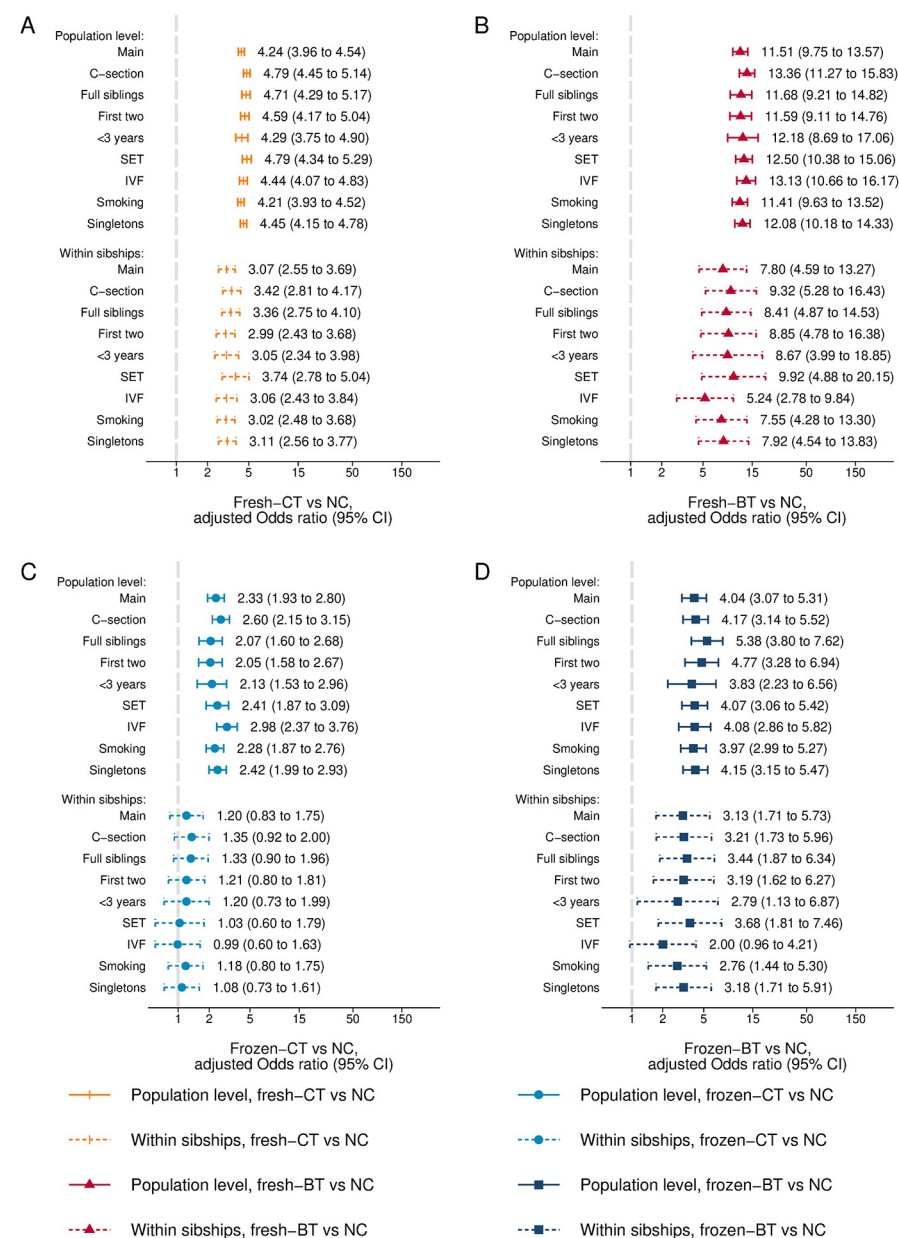

**Fig 3. Adjusted odds ratio of placenta previa after (A) fresh-CT, (B) fresh-BT, (C) frozen-CT, and (D) frozen-BT compared to NC in sample 3, at the population level and within mothers.** Fresh-CT, fresh cleavage stage embryo transfer; NC, natural conception; fresh-BT, fresh blastocyst transfer; frozen-CT, frozen cleavage stage embryo transfer; frozen-BT, frozen blastocyst transfer. Adjusted for year of delivery (categorical: 1988–1996, 1997–2001, 2002–2006, 2007–2011, 2012–2015), maternal age (categorical: 20–24, 25–29, 30–34, 35–39, 40–45), parity (categorical: 0, 1, 2, 3), and country (only included in analyses at the population level because it is constant within mothers). Analyses in "Smoking" are additionally adjusted for smoking status in the subpopulation where this information was available. In "C-section" the outcome of placenta previa is restricted to pregnancies delivered by cesarean section, whereas "Full siblings," "First two," "<3 years," "SET," "IVF," and "Singletons" refer to restriction of analyses to full siblings (i.e., same mother and father), each mothers' first 2 consecutive deliveries, only consecutive deliveries less than 3 years apart, single embryo transfer (using all deliveries after NC as reference), in vitro fertilization (using all deliveries after NC as reference), and singleton pregnancies, respectively.

In line with our results, Spangmose and colleagues, Korosec and colleagues, Volodarsky-Perel and colleagues, and Cirillo and colleagues reported risks of 4.1%, 4.0%, 3.5%, and 4.3%, respectively, of placenta previa after fresh-BT, from a total of 4,601, 802, 425, and 202 pregnancies, respectively [23–26]. The study by Spangmose and colleagues is also based on the CoN-ARTaS cohort [23]. We shed new light on those findings by controlling for stable maternal factors and by improving our understanding of the potential contribution from different infertility factors.

A nationwide Japanese study of single embryo transfers (*n* = 48,158) showed no clear differences across combinations of culture duration and cryopreservation but had an appreciably lower risk of placenta previa after ART-conception overall (0.7% to 0.9%) than in our study [27].

Endometriosis was strongly associated with placenta previa in ART-conceived pregnancies, consistent with previous comparisons of ART-conceiving women with endometriosis to either ART-conceiving women without endometriosis [18,30,41–43], or women who conceived naturally with or without endometriosis [29,42]. We elaborate these observations by using within-mother analyses to disentangle the contribution of ART from that of endometriosis. Our results indicate that ART is associated with a doubled odds of placenta previa in women with endometriosis, from an already higher baseline risk than among women with other infertility factors.

In ART, PCOS was associated with lower risk of placenta previa than other infertility factors. This is consistent with previous studies comparing women using ART for ovulation disorders to women with NC [29] and to women without ovulation disorders using ART [18,30]. In contrast to previous studies, we studied PCOS instead of ovulation disorders, because the latter also includes poor ovarian reserve due to advanced maternal age, which makes interpretation more challenging.

Several mechanisms for a higher risk of placenta previa after ART-conception have been suggested. One hypothesis is that the position of the catheter during embryo transfer might play a role [44], supported by observations that transfer site corresponds to implantation site in 80% of transfers [45]. Another hypothesis is that the catheter sometimes causes uterine contractions when passing through the cervix, which may displace the embryo [46]. Enhanced uterine peristalsis has also been hypothesized to play a role in the higher risk of previa in women with endometriosis [47].

Fresh-ET is associated with higher levels of estradiol and progesterone at the time of embryo transfer compared to frozen-ET, as a result of ovarian stimulation [48,49]. Increased levels of estradiol have been proposed to lower the threshold for uterine contractions in response to mechanical stimuli, which could explain the higher risk of placenta previa after fresh-ET [46]. However, a later study found no association between peak estradiol during ovarian stimulation and subsequent risk of placenta previa after fresh-ET [47]. Further, a premature rise in progesterone levels has been shown to induce premature maturation of the endometrium [49]. It seems conceivable that a longer secretory phase might make the endometrium in the lower segment of the uterus more receptive to embryo implantation in fresh-ET compared to frozen-ET and NC. This effect may be enhanced in fresh-BT as extended culture duration results in prolonged maturation of the endometrium before embryo transfer. Indeed, thicker endometrium is associated with higher risk of placenta previa after fresh-ET [47,50,51], but not after frozen-ET [52,53]. A greater number of retrieved oocytes have been associated with higher progesterone levels [54]. We had no information on progesterone levels during ovarian stimulation, but a study of deliveries after fresh-ET in Sweden between 2002 and 2015 found that retrieval of more oocytes was associated with higher risk of placenta previa in singleton pregnancies [55].

Future studies should explore the association between fresh-ET, premature rise in progesterone levels and subsequent risk of placenta previa. In addition, analyses of the risk of placenta previa according to embryo culture duration and cryopreservation in women with endometriosis, and women with other infertility factors, is warranted.

## Strengths and limitations

A key strength of this study is the within-mother analyses where each woman serves as her own control [56]. In a Norwegian study population that partly overlaps with ours, Romundstad and colleagues also found higher risk of placenta previa after ART compared to NC using within-mother analyses [44]. We provide new insights by deepening our understanding of the potential roles of different ART methods and infertility factors.

Given the observational nature of this study, we cannot exclude the possibility of unknown or unmeasured confounding. While we expect within-mother analyses to be less confounded from stable maternal factors (such as cause of infertility and genetic predisposition for placenta previa), we cannot rule out confounding from non-stable maternal factors. However, we note that the risk of placenta previa was highest after fresh-ET regardless of delivery order among women who gave birth after different conception methods in their first 2 consecutive births.

We had limited data on ART treatment details, which was missing for all Finnish ART-pregnancies. Furthermore, cryopreservation and blastocyst transfers were uncommon in the first half of our study period [23,33]. Consequently, there were not enough pregnancies after fresh-BT and frozen-BT to do country-specific analyses. Instead, we included country as a categorical covariate in the regression models to adjust for differences between countries. Similarly, we did not have sufficient statistical power to explore the risk of placenta previa after fresh cleavage stage embryo transfer (fresh-CT), frozen-CT, fresh-BT, and frozen-BT according to infertility factors.

While we cannot rule out information bias from differences in data sources and registration practice of placenta previa, we expect this to be non-differential with respect to conception method. Furthermore, we assume negligible information bias from defining pregnancies after ovulation induction and insemination as natural conceptions, because these pregnancies are very few compared to the remaining background population. Moreover, beyond restricting the diagnosis of placenta previa to deliveries with cesarean section in sensitivity analysis, we could not separate a total, partial, or marginal placenta previa from a low-lying placenta. However, low-lying placentas also appear clinically relevant [57,58], and their management are similar to placenta previa [59].

The higher risk of placenta previa in women with endometriosis in our study might not reflect the risk associated with endometriosis in general, since we restricted diagnosis of endometriosis to women with delivery after ART. Of note, an ICD-code of endometriosis in the Swedish National Patient Registry showed a positive predictive value of 97.8% compared to the gold standard of reviewing patient charts [60]. Furthermore, our assignment of women to specific infertility factor groups will not always reflect the main cause of infertility, which can also be difficult to ascertain clinically [61]. We had no data on the severity or diagnostic basis of endometriosis, both of which may be associated with the risk of placenta previa [62,63].

A strength of this study was the consistency of results across a series of sensitivity analyses. The population-based design with inclusion of both singleton and multifetal pregnancies is expected to better capture the total effect of ART compared to previous studies [21–27]. ART treatment as well as the ART-treated populations continues to change, thus, there is a possibility that our findings of a higher risk of placenta previa after ART might not apply to women receiving treatment today. Although the newest data in this study were from 2015, we expect

the findings to apply to contemporary treatment practices such as single embryo transfer and blastocyst culture. Fresh-BT now constitutes 18% to 21% of all Swedish embryo transfers, and frozen-BT has become far more prevalent than frozen-CT [64]. Indeed, a previous study on the CoNARTaS cohort showed a steadily increasing risk of placenta previa after ART compared to NC between 1988 and 2015, which might partly be explained by a shift from cleavage stage to blastocyst embryo transfer [10]. Nevertheless, caution is needed when generalizing our findings to populations outside the Nordic countries, as no clear pattern between placenta previa, embryo culture duration and cryopreservation status was found in a large registry-based study from Japan [27]. Furthermore, results from within-mother analyses might not generalize to all women using ART, as these analyses are restricted to multiparous women with different conception methods.

## Conclusions

Our study indicates that the higher risk of placenta previa after ART is associated with both treatment and maternal factors, where fresh-ET, and especially fresh-BT, was associated with an increased risk of placenta previa. Furthermore, women diagnosed with endometriosis and receiving ART seemed to be at particularly high risk, possibly due to contributions from ART. Further insights into the mechanisms responsible might lead to development of safer ART methods. Although the risk of placenta previa may not be weighted heavily when choosing the type of ART treatment, it might be considered along with the risk of other complications, especially if the alternative treatment types will not reduce the chances of achieving a pregnancy.

## Supporting information

**S1 Checklist. The RECORD statement—checklist of items, extended from the STROBE statement that should be reported in observational studies using routinely collected health data.**
(DOCX)

**S1 Table. Data sources and registration practice for placenta previa in the Nordic countries during the study period.**
(DOCX)

**S2 Table. Data sources and registration practice for infertility factors in the Nordic countries during the study period.**
(DOCX)

**S3 Table. ICD codes used for identification of infertility factors in the Nordic countries during the study period.**
(DOCX)

**S4 Table. Observed risk and adjusted odds ratio of placenta previa in second pregnancy according to conception method and cesarean section in first pregnancy in sample 1.**
(DOCX)

**S5 Table. Observed risk and adjusted odds ratio of placenta previa in second pregnancy according to conception method and placenta previa in first pregnancy in sample 1.**
(DOCX)

**S6 Table. Description of pregnancies with and without placenta previa according to conception method and plurality in sample 1.**
(DOCX)

**S1 Text. More detailed description of sensitivity analyses.**
(DOCX)

**S1 Study Protocol. Risk of placenta previa in assisted reproductive technology: A Nordic registry-based population study with within-sibship analyses.**
(PDF)

## Acknowledgments

We would like to thank the staff in the ART clinics and labor departments in all 4 countries for their efforts in recording the data used in this study.

## Author Contributions

**Conceptualization:** Eirik Landsverk, Kjersti Westvik-Johari, Ulla-Britt Wennerholm, Christina Bergh, Frederik Kyhl, Anne Lærke Spangmose, Ditte Vassard, Anja Pinborg, Kristiina Rönö, Mika Gissler, Sindre Hoff Petersen, Signe Opdahl.

**Data curation:** Eirik Landsverk.

**Formal analysis:** Eirik Landsverk.

**Funding acquisition:** Ulla-Britt Wennerholm, Christina Bergh, Anja Pinborg, Mika Gissler, Signe Opdahl.

**Investigation:** Ulla-Britt Wennerholm, Christina Bergh, Anja Pinborg, Mika Gissler, Signe Opdahl.

**Methodology:** Eirik Landsverk, Kjersti Westvik-Johari, Ulla-Britt Wennerholm, Christina Bergh, Frederik Kyhl, Anne Lærke Spangmose, Ditte Vassard, Anja Pinborg, Kristiina Rönö, Mika Gissler, Sindre Hoff Petersen, Signe Opdahl.

**Project administration:** Signe Opdahl.

**Resources:** Kjersti Westvik-Johari, Signe Opdahl.

**Supervision:** Kjersti Westvik-Johari, Signe Opdahl.

**Visualization:** Eirik Landsverk.

**Writing – original draft:** Eirik Landsverk.

**Writing – review & editing:** Eirik Landsverk, Kjersti Westvik-Johari, Ulla-Britt Wennerholm, Christina Bergh, Frederik Kyhl, Anne Lærke Spangmose, Ditte Vassard, Anja Pinborg, Kristiina Rönö, Mika Gissler, Sindre Hoff Petersen, Signe Opdahl.

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
