## [Editor Report · Decision Letter 0]

14 Aug 2024

Dear Dr Landsverk, 

Thank you for submitting your manuscript entitled "Risk of placenta previa in assisted reproductive technology: A Nordic population study with sibling analyses" for consideration by PLOS Medicine.

Your manuscript has now been evaluated by the PLOS Medicine editorial staff and I am writing to let you know that we would like to send your submission out for external peer review.

Please re-submit your manuscript within two working days, i.e. by Aug 16 2024 11:59PM.

Feel free to email me at lgaynor@plos.org if you have any queries relating to your submission.

Kind regards,

Louise Gaynor-Brook, MBBS PhD

Senior Editor

PLOS Medicine

---

## [Decision Letter · Decision Letter 1]

28 Nov 2024

Dear Dr Landsverk,

Many thanks for submitting your manuscript "Risk of placenta previa in assisted reproductive technology: A Nordic population study with sibling analyses" (PMEDICINE-D-24-02651R1) to PLOS Medicine. The paper has been reviewed by subject experts and a statistician; their comments are included below and can also be accessed here: [LINK]

After discussing the paper with the editorial team and an academic editor with relevant expertise, I'm pleased to invite you to revise the paper in response to the reviewers' comments. We plan to send the revised paper to some or all of the original reviewers, and we cannot provide any guarantees at this stage regarding publication.

We ask that you submit your revision by Dec 19 2024 11:59PM. However, if this deadline is not feasible, please contact me by email, and we can discuss a suitable alternative.

Don't hesitate to contact me directly with any questions (lgaynor@plos.org). 

Best regards, 

Louise 

Louise Gaynor-Brook, MBBS PhD 

Senior Editor

PLOS Medicine

lgaynor@plos.org

Comments from the academic editor:

I agree that the fact the data goes back a long way is a significant limitation: this, and the potential impact of changing ART techniques, could be discussed in more depth. I am not convinced of the need for a country stratified analysis.

Comments from the reviewers: 

Reviewer #1: In this paper, the authors conduct a retrospective registry study examining the relative odds of placenta previa in pregnancies using assisted retroductive technology (ART) vs. natural conception (NC). Major strengths of this study include the large sample size derived from a set of high quality national registries, which allows for the ability/power to examine specific sub-populations and sensitivity comparisons, and the use of a within-mother analysis in order to reduce heterogeneity due to mother-specific confounders. In general, we commend the authors on this well conducted study. A few suggestions are provided below.

I actually thought that Figure 2 regarding the nine potential combinations of first vs. second delivery modality deserved much more discussion than was provided in the manuscript. Some important patterns: we see lower predicted risk of placenta previa between the first delivery and second delivery when the first delivery is fresh-ET /except/ when the second delivery is also fresh-ET. Additionally, the magnitude of the increased predicted risk for first-delivery-NC pairs seems to vary greatly depending on the modality of the second delivery. These "differences in differences" so to speak are potentially of clinical interest, though they are only observational associations found in the data. 

Throughout the manuscript, there is occasional use of unsubstantiated causal language, which is not acceptable given the observational nature of the study (e.g., line 233-234: "ART increased the risk" as opposed to "ART was associated with increased risk beyond..."). One major area where we believe this language should definitely be changed is in line 71-72 ("assisted reproductive technology contributed to the increased risk"), which is a fully causal statement that was not supported by the analysis (the authors did not conduct a causal inference-informed approach). Despite the sibling design and the within-mother matching, and the comments on lines 288-290, such statements are still not appropriate. For instance, there still may remain potential unmeasured confounding due to time-varying factors between pregnancies. Thus, in the conclusion lines 319-320, it is also inappropriate to state that the higher observed risk is "attributed" to a combination of treatment and maternal factors (even if there is a biologically plausible explanation). To be most appropriate, we would prefer language such as "the highest risk of placenta previa after ART is associated with both...". For some more minor examples of problematic causal language, it may be more appropriate in lines 69-70 to state "The largest difference in risk was seen" as opposed to "The highest risk increase," as the latter (current) language implies a causal relationship. Along these lines, it may be worth stating that the confounders referred to in lines 157-159 are /previously known confounders/ (with the implication being that there may always be unmeasured confounders, which will certainly be the case in large registry studies such as this one, e.g., environmental exposures, etc. 

Although the authors mention that a country-stratified analysis was not performed, it still may be of interest, perhaps due to potentially different medical practices or circumstances between (admittedly very similar) countries; even in a simple additionally sensitivity analysis that simply incorporated country as a categorical variable to be adjusted for, is there truly no evidence of difference across country lines? The inclusion of this variable is unlikely to affect inferential results if the relationship is truly null given the large sample size; it struck me as a strange modeling choice not to use data so immediately available and that may be potentially worth exploring. On a somewhat related note, I appreciated your discussion of the generalizability of the study; I would have preferred if you were even more explicit in mentioning that Nordic countries may not have populations which are comparable to those of other countries. For instance, you do discuss the difference in overall rate of placenta previa with respect to Ishihara et al., but it would have been stronger to explicitly mention this potential drawback (i.e., generalizing to other populations) as well. 

Finally, as a very small curiosity of mine, I wonder whether parity was included as a linear or categorical variable in your models when adjusting for them. Some additional clarity there would be appreciated. 

In any case, a well-written, clear, and well-conducted study. It is likely to be of great interest to epidemiologists and practitioners. 

Reviewer #2: This is an excellent study performed by a highly experienced group of investigators, using a combined Nordic dataset that has been used previously. The study question was whether ART increases the risk for placenta previa. The authors evaluate this in a full population of ART and Naturally Conceived (NC) pregnancies as well as sequential pregnancies within the same woman. They analyze over 5 million pregnancies in 3 million mothers. They find that ART increases the rate of placenta previa and that specific ART techniques including fresh transfer (compared to frozen) and blastocyst transfer (compared to cleavage stage) showed the highest rates. In addition, specific infertility diagnoses were investigated and showed that endometriosis patients had a higher rate of placenta previa.

The data analysis in the paper is exhaustive. The tables and figures are appropriate, and the reference list appears comprehensive.

I had one question about the study population as described in Methods. It was not clear to me (perhaps I missed it), what was done with more than 2 pregnancies per woman. I assume that all pregnancies were included in the "Eligible sample". For the within woman comparisons, were just the first 2 pregnancies compared or was pregnancy 1 compared with pregnancy 2 and then pregnancy 2 compared with pregnancy 3? This should be clarified. Also, was any calculation done to adjust for multiple pregnancies to the same woman in the overall analysis?

I think that the authors should add the limitation that the study period spanned the years 1988-2015 which means it includes very old cases and that it does not include the most recent cases. While year was adjusted for, there have been many changes to ART protocols over the years that could make a difference.

Reviewer #3: Congratulations on a very substantial body of work establishing that it is likely that presented previous is more common in IRT with variations dependent on the cause of infertility the staging of embryo transfer and particular interest presence of endometriosis. As much as I appreciate the statistical value of odds ratios, as a clinician the raw numbers are probably more relevant when counselling patients. I would ask that the the final sentence in the conclusion is unnecessary. It perhaps should be

From a clinician's perspective, the increased risks of placenta praevia, which is rarely a lifethreatening complication, are unlikely to change the chosen ART treatment approach. The primary goal of maximum pregnancy success will override these statistical findings. section rates but against a background of high rates after ART , the extra numbers will be relatively small .

* Please upload any figures associated with your paper as individual TIF or EPS files with 300dpi resolution at resubmission; please read our figure guidelines for more information on our requirements: http://journals.plos.org/plosmedicine/s/figures. While revising your submission, please upload your figure files to the PACE digital diagnostic tool, https://pacev2.apexcovantage.com/. PACE helps ensure that figures meet PLOS requirements. To use PACE, you must first register as a user. Then, login and navigate to the UPLOAD tab, where you will find detailed instructions on how to use the tool. If you encounter any issues or have any questions when using PACE, please email us at PLOSMedicine@plos.org.

FIGURES AND TABLES

SUPPLEMENTARY MATERIAL

REFERENCES

STUDY TYPE-SPECIFIC REQUESTS: OBSERVATIONAL STUDIES

* Abstract: Please include the study design, population and setting, number of participants, years during which the study took place (enrollment and follow up), length of follow up, and main outcome measures.

* Please ensure that the study is reported according to the RECORD guideline (available from https://www.record-statement.org) and include the completed checklist as Supporting Information. Please add the following statement, or similar, to the Methods: "This study is reported as per the Reporting of Studies Conducted using Observational Routinely-Collected Data (RECORD) guideline (S1 Checklist)." When completing the checklist, please use section and paragraph numbers, rather than page numbers.

* For all observational studies, in the manuscript text, please indicate: (1) the specific hypotheses you intended to test, (2) the analytical methods by which you planned to test them, (3) the analyses you actually performed, and (4) when reported analyses differ from those that were planned, transparent explanations for differences that affect the reliability of the study's results. If a reported analysis was performed based on an interesting but unanticipated pattern in the data, please be clear that the analysis was data driven. 

* Please state in the Methods section whether the study had a prospective protocol or analysis plan. If a prospective analysis plan (from your funding proposal, IRB or other ethics committee submission, study protocol, or other planning document written before analyzing the data) was used in designing the study, please include the relevant document(s) with your revised manuscript as a Supporting Information file to be published alongside your study and cite it in the Methods section. A legend for this file should be included at the end of your manuscript. If no such document exists, please make sure that the Methods section transparently describes when analyses were planned, and when/why any data-driven changes to analyses took place. Changes in the analysis, including those made in response to peer review comments, should be identified as such in the Methods section of the paper, with rationale.

---

## [Decision Letter · Decision Letter 2]

10 Jan 2025

Dear Dr. Landsverk,

Thank you very much for re-submitting your manuscript "Risk of placenta previa in assisted reproductive technology: A Nordic population study with sibling analyses" (PMEDICINE-D-24-02651R2) for review by PLOS Medicine. I am writing on behalf of my colleague Louise Gaynor Brook, who is away from the office.

I have discussed the paper with my colleagues and the academic editor and it was also seen again by one of the original reviewers. I am pleased to say that provided the remaining editorial and production issues are dealt with we are planning to accept the paper for publication in the journal.

The remaining issues that need to be addressed are listed at the end of this email and minor edits are in the attached word document. Any accompanying reviewer attachments can be seen via the link below. Please take these into account before resubmitting your manuscript:

[LINK]

We look forward to receiving the revised manuscript by Jan 17 2025 11:59PM.   

Sincerely,

Alison Farrell PhD

Senior Editor 

PLOS Medicine

plosmedicine.org

Requests from Editors:

Please note edits in the attached file. 

Please be sure to acknowledge unmeasured confounders in your discussion of study limitations.

Please reiterate that while country-specific analyses were not feasible in all instances due to missing data, that country was included as a categorical covariate in analyses.

Please state early in the text that you followed a prespecified statistical analysis plan.

If new code was generated in this study, please deposit in GitHub and provide the url.

Please address the following issues:

Data availability statement is missing. For each data source used in your study: 

Author contribution statement is missing.

Acknowledgements section is blank (acknowledge participants whose data are used?)

Clarify use of data--ethics/consent waived? Clarify how identities are protected given that you have used national identity numbers.

GATHER statement: If appropriate, please report your data according to GATHER and enclose a completed GATHER checklist as a supplementary document. See http://gather-statement.org/ In the checklist please include sufficient text excerpted from the manuscript to explain how you accomplished all applicable items. 

Competing Interests Statement: * All authors must declare their relevant competing interests per the PLOS policy, which can be seen here: https://journals.plos.org/plosmedicine/s/competing-interests For authors with ties to industry, please indicate whether any of the interests has a financial stake in the results of the current study. 

Comments from Reviewers:

Reviewer #1: Thank you for comprehensively addressing the points raised in my original review. We are satisfied with the changes and have no further comments.

[LINK]

---

## [Editor Report · Decision Letter 3]

20 Jan 2025

Dear Dr Landsverk, 

On behalf of my colleagues and the Academic Editor, Jenny Myers, I am pleased to inform you that we have agreed to publish your manuscript "Risk of placenta previa in assisted reproductive technology: A Nordic population study with sibling analyses" (PMEDICINE-D-24-02651R3) in PLOS Medicine.

PRESS

Sincerely, 

Alison Farrell, PhD 

Senior Editor 

PLOS Medicine